# Improving the Clinical Utility of Platelet Count for Cancer Detection in Primary Care: A Cohort Study in England, Canada, and Australia

**DOI:** 10.3390/cancers16173074

**Published:** 2024-09-04

**Authors:** Luke T. A. Mounce, Raff Calitri, Willie Hamilton, Meena Rafiq, Jon D. Emery, Vasily Giannakeas, Joanne Kotsopoulos, Sarah E. R. Bailey

**Affiliations:** 1University of Exeter Medical School, University of Exeter, Exeter EX1 3DB, UK; l.t.a.mounce@exeter.ac.uk (L.T.A.M.); r.calitri@exeter.ac.uk (R.C.); w.hamilton@exeter.ac.uk (W.H.); 2Department of General Practice and Primary Care, Centre for Cancer Research, University of Melbourne, Grattan Street, Parkville, VIC 3010, Australia; meena.rafiq@unimelb.edu.au (M.R.); jon.emery@unimelb.edu.au (J.D.E.); 3Epidemiology of Cancer Healthcare & Outcomes (ECHO) Group, Department of Behavioural Science, Institute of Epidemiology and Health Care (IEHC), UCL, London WC1E 7HB, UK; 4Research and Innovation Institute, Women’s College Hospital, Toronto, ON M5S 1B2, Canada; vasily.giannakeas@wchospital.ca (V.G.); joanne.kotsopoulos@wchospital.ca (J.K.); 5ICES, Toronto, ON M5T 3M6, Canada; 6Department of Epidemiology, Dalla Lana School of Public Health, University of Toronto, Toronto, ON M5T 3M7, Canada

**Keywords:** cancer, diagnosis, primary care, family medicine, full blood count, platelet count

## Abstract

**Simple Summary:**

The platelet count is an established marker of cancer. In healthy populations, platelet count varies by age and sex; despite that variation, a single reference range is used. This study aimed to identify age- and sex-specific upper thresholds for platelet count at which cancer should be considered in primary care, using audits of primary care-based clinical data in England, Canada, and Australia. Across all three cohorts, there was a clear upwards trend in cancer incidence with increasing platelet count for both sexes and at all age groups. The appropriate threshold will vary by country and will depend on local healthcare needs and priorities. As colorectal and lung cancers predominate, initial investigation may target these sites. Further investigation, depending on the patient’s symptoms, could be CT imaging, endoscopy, or CA125.

**Abstract:**

The platelet count, a component of the full blood count, has been identified as a useful diagnostic marker for cancer in primary care. The reference range for the platelet count is 150 to 400 or 450 × 10^9^/L; this range does not account for natural variation in platelet count by age and sex. This study used three primary care cohorts from England, Canada, and Australia. Patients aged 40 years and over with a full blood count were included and stratified by age (in 10-year bands), sex, (male/female), and platelet count group. Cancer incidence within one year of the test date was estimated from linked registry data. In all three countries, there was a clear upwards trend in cancer incidence with increasing platelet count for both sexes and at all age groups. Lung and colorectal were the most common sites. These results have important implications for the international application of this work; analysis of local health datasets will be crucial to determining appropriate thresholds. Appropriate upper thresholds will depend on local populations, healthcare needs, and priorities. Further research is needed to assess the likely impact of new recommendations on the healthcare system, on cancer outcomes, and patient benefit.

## 1. Introduction

The platelet count, a component of the full blood count, has been identified as a useful diagnostic marker for cancer in patients presenting to primary care. The standard reference range for the platelet count in the UK is 150 to 400 or 450 × 10^9^/L [1], irrespective of the age or sex of the patient; we know that platelet count varies by age and sex in healthy populations [2,3,4,5].

A cohort study based in English primary care from 2000 to 2014 found that 11% of males and 6% of females aged 40 years and over with thrombocytosis (platelet count > 400 × 10^9^/L) were diagnosed with cancer within one year of that test result [6]. Colorectal and lung cancer were the most common cancer types. These findings were recently supported by a cohort study in Ontario, Canada, which found that 6% of males and 4% of females aged 40 to 75 years with thrombocytosis were diagnosed with a solid cancer within one year of the blood test [7]. The lower percentages in the latter are likely to reflect the upper age cut-off (i.e., 75 years) applied in the Canadian dataset. A second study set in English primary care of 295,312 patients with a normal platelet count (150 to 400 × 10^9^/L) from 2005 to 2014 found that males aged 60 years and over with platelet counts of 376 to 400 × 10^9^/L had one-year cancer incidence rates greater than 3% [8].

Collectively, these studies demonstrate that cancer incidence rates in patients with raised platelets exceed the 3% risk threshold set by the UK National Institute for Health and Care Excellence (NICE) for investigation for suspected cancer (guidance NG12). In the 2015 update of NG12, thrombocytosis was included as a feature of lung and endometrial cancer, warranting further investigation [9].

Given that the 3% risk level set by NICE for the investigation of a suspected cancer may be achieved at different platelet count levels in males and females and at different ages, age- and sex-specific thresholds could take advantage of that natural variation to provide personalised estimates of cancer risk. Age- and sex-specific reference ranges have been proposed [2,10] but have not been adopted in UK health guidance.

This study aimed to identify age- and sex-specific upper thresholds for platelet count at which cancer should be considered in primary care, using audits of primary care-based clinical data in England, Canada, and Australia. The objective was to estimate one-year cancer incidence in patients with a primary care-ordered full blood count, stratified by age, sex, and platelet count in these three countries.

## 2. Materials and Methods

Three retrospective cohort studies were set in primary care in England, Canada, and Australia. The study set in England used routine electronic medical records of primary care consultations from the Clinical Practice Research Datalink Aurum (CPRD Aurum) [11] with National Cancer Registration and Analysis Service (NCRAS) linkage. Patients with a full blood count from 1 January 2016 to 31 December 2017 were included. Follow-up cancer registration data from the National Cancer Registration and Analysis Service (NCRAS) were available until 31 December 2018. All patient medical encounters, demographics, diagnoses, symptoms, and test results within the study period were recorded in the dataset using a series of unique numeric identifiers.

The study set in Ontario, Canada, used electronic health care databases from provincial laboratory records and the provincial cancer registry. Datasets were linked using unique encoded identifiers and analysed at ICES (formerly the Institute for Clinical Evaluative Sciences), a non-profit research institute whose legal status under Ontario’s health information privacy law allows it to collect and analyse health care and demographic data, without individual consent, for health system evaluation and improvement. The Ontario Laboratory Information System (OLIS) database was used to identify provincial residents with a platelet count record between 1 January 2016 and 31 December 2017. Included residents were linked to the Ontario Cancer Registry to capture incident cancers that occurred historically and in a 12-month follow-up period.

The study set in Australia used routinely collected primary care data from the Patron primary care dataset [12], which contains de-identified clinical and administrative data from over 130 general practice (GP) clinics across the state of Victoria, including GP blood test results electronically transmitted into the electronic records from laboratories [13]. This dataset was linked through the Centre for Victorian Data Linkage (CVDL) [14] to the Victorian Admitted Episodes Dataset (VAED), which contains details of diagnoses from all Victorian hospital admissions, and the Victorian Cancer Registry (VCR) [15]. Linked patients with a primary care full blood count result from 20 July 2007 to 20 July 2021 were included. Follow up cancer registration data from VCR were available until 20 July 2022. An expanded study timeframe was used to maximise sample sizes. Data were extracted on patient demographics, full blood count (date and result), and month of cancer diagnosis.

All three studies included patients aged 40 years and over with a primary care-ordered blood test in the eligible time frame; the first blood test on record for each patient in the study period was taken as the index test. Patients were excluded if they had a record of a cancer diagnosis prior to index test date. In the Australian study, for bowel and breast cancer, patients were excluded if their cancer within one year of index date was recorded as screen detected in the VCR.

The primary exposure was platelet count (units). The primary outcome was a record of an incident cancer diagnosis within one year of index test date, of any type except non-melanoma skin cancers (as is standard practice in incidence studies) and platelet malignancies (to ensure results reflect an unexplained increased platelet count as a risk marker for cancer). The date of the first record of cancer in the patient notes after index test date was taken as the date of diagnosis. The site associated with that record was considered the primary site of diagnosis. If patients had multiple cancers diagnosed on the same day, the following procedure was followed: (1) duplicate records of cancer were dropped; (2) the staged cancer was accepted as the primary site; (3) the most advanced stage cancer was accepted if multiple staged cancers were recorded; and (4) if there were multiple staged cancers from different sites, one was selected at random.

Patients were stratified by sex (male/female, as recorded in the patient records), and then by age and platelet count as follows:

By age, into five groups: aged 40 to 49 years; aged 50 to 59 years; aged 60 to 69 years; aged 70 to 79 years; aged 80 years and over.

By platelet count, into six groups: under 350 × 10^9^/L; 350 to 399 × 10^9^/L; 400 to 449 × 10^9^/L; 450 to 499 × 10^9^/L; 500 to 549 × 10^9^/L; 550 × 10^9^/L and over.

One-year cancer incidence (percentage of patients in each sex/age/platelet count strata) was calculated with 95% confidence intervals (CIs) using the Clopper–Pearson exact method. We identified the platelet count at which one-year cancer incidence exceeded 3% (the threshold set by NICE for urgent investigation for suspected cancer) and 7% (this represents the 3% threshold set by NICE [9], plus the 4% artefactual increased risk of cancer in patients being selected for a blood test in primary care [6].

The analysis was conducted with Stata MP version 14.0. R Studio version 22.07.1 + 554 was used to produce Figure 1 with the ggplot2 package.

## 3. Results

Cancer incidences by sex, age group, and platelet count group for the three cohorts are displayed in Figure 1.

### 3.1. English Cohort

There were 3,087,436 patients meeting eligibility criteria with recorded platelet counts in the CPRD between 1 January 2016 and 31 December 2017. The patient demographics are described in Table 1. The mean patient age was 62.59 years (SD 13.75). Over half of the cohort was female (*n* = 1,720,609, 55.73%). The platelet count thresholds that correspond to a 7% or higher cancer incidence were: for males aged 60 to 69 years: 450 × 10^9^/L; for males aged 70+ years: 350 × 10^9^/L; for females aged 60 to 69 years: 550 × 10^9^/L; for females aged 70+: 500 × 10^9^/L (Figure 1). The five most common cancer sites diagnosed in these patients exceeding the 7% threshold were lung, colon, prostate, rectum, and kidney. The ten most common cancers in the full cohort are displayed in Table 2 (by cohort).

### 3.2. Canadian Cohort

There were 3,140,451 eligible patients with a platelet count record in the OLIS database between 1 January 2016 and 31 December 2017. The mean patient age was 60.8 (SD 12.6). Over half of the cohort was female (*n* = 1,747,701, 55.7%). The platelet count thresholds that correspond to a 7% cancer incidence were: for males aged 70 to 79 years: 400 × 10^9^/L; for males aged 80+: 350 × 10^9^/L; for females aged 80+ years: 500 × 10^9^/L. The most common sites of malignancy were colon, lung, prostate, lymphoma, and breast.

### 3.3. Australian Cohort

There were 266,900 recorded eligible platelet counts in the Patron dataset between 20 July 2007 and 20 July 2021. The mean patient age was 57.38 (SD 13.88). Over half of the cohort was female (*n* = 149,210, 55.9%). Power to detect differences to our thresholds was much reduced in this smaller sample. Observed cancer incidence significantly exceeded the 7% threshold in males aged 60+ with counts of 550 × 10^9^/L or greater. Broadly, the observed incidences trended in line with the two larger cohorts. The most common sites of malignancy were colon, haematological, lung, prostate, and rectum.

**Table 2 cancers-16-03074-t002:** One-year post-test cancer incidence by site in the three cohorts.

	English Cohort N = 3,087,436	Canadian Cohort N = 3,140,451	Australian Cohort N = 266,900
Cancer incidence, *n* (%)	58,001 (1.88)	48,156 (1.53)	152 (0.05)
Ten * most common tumours, *n* (% of tumours)	Prostate; 9417 (16.24) Lung; 7611 (13.12) Breast; 5547 (9.56) Colon; 5324 (9.18) Lymphoma; 2778 (4.79) Rectum; 2527 (4.36) Pancreas; 2266 (3.91) Kidney; 1899 (3.27) Leukaemia; 1841 (3.17) Bladder; 1776 (3.06)	Prostate; 6509 (13.52) Breast; 5660 (11.75) Lung; 5532 (11.49) Colon; 4941 (10.26) Lymphoma; 3472 (7.21) Bladder; 2298 (4.77) Melanoma; 1808 (3.75) Endometrial; 1707 (3.54) Kidney; 1401 (2.91) Cervix; 1382 (2.87)	Colon; 36 (23.68) Lung; 27 (17.6) Prostate; 8 (5.26) Rectum; 8 (5.26)

* In the Australian cohort, tumours with counts <5 are redacted to preserve anonymity of patients.

## 4. Discussion

Across all three cohorts, there was a clear upwards trend in cancer incidence with increasing platelet count for both sexes and at all age groups. Increases in cancer incidence with age were observed across all groups, although absolute risk was consistently lower in females than males across platelet groups. This is likely to be due to higher “baseline” platelet counts in females [2]. The observed trends in cancer incidence were consistent across the three cohorts, with very comparable results between England and Ontario and wider confidence intervals in the Australian results due to the much smaller sample size. The Canadian and Australian cohorts were younger than the English cohort, which would have resulted in a slightly lower overall incidence. Taking a 7% risk threshold approach would result in different upper thresholds being applied across the three nations. For example, at a 7% threshold, males in their 60s would qualify for suspected cancer investigation in the English cohort, but not in the cohort from Ontario (although they were close to this threshold). This may reflect differences in testing practices.

## 5. Strengths and Limitations

This study included two large cohorts from England and Ontario, giving highly reliable estimates of cancer incidence. The Australian cohort, although smaller, supports the findings in the other two countries. Despite this consistency, there are likely to be some differences between the three populations. None of the datasets have recorded reasons for blood testing; there will be patients in each cohort who reported high-risk symptoms for cancer, which would have prompted referral, where a suspiciously raised platelet count would not have changed the clinical action. Previous studies of English primary care data have found that one third of patients with lung and colorectal cancer who had pre-diagnostic thrombocytosis had no high-risk symptoms recorded the year before their diagnosis [6]. Previous studies in Ontario found a significant decline in the association of platelet count and cancer incidence beyond one year from the blood test.

We benchmarked our results to the UK’s threshold of a 3% risk of cancer warranting urgent investigation, with an added 4% increase to represent the fact that those having blood tests during medical care are a selected group with a higher pooled incidence of cancer. Both thresholds—3% and 7%—are semi-arbitrary, though the 3% figure has gathered considerable traction in the UK. Our results can be used with different thresholds, with patients generally seeking investigation even if the risk of cancer is much lower [16]. The appropriate threshold will vary by country; in the present study, there may be a lower “artificially increased risk” associated with being selected for a blood test in Ontario and Australia if GPs have lower thresholds for ordering tests compared to in England (anecdotally, the authors consider this to be the case). Differences in baseline risk of cancer in the three cohorts will also impact the results of this study. In a study set in a hospital in Italy, 83 of 917 consecutive adult admissions (9%) had thrombocytosis using the traditional reference range; 12.0% of these were diagnosed with cancer [17]. When applying personalised reference ranges (stratified by age and sex), the number of patients with platelets over the upper threshold increased to from 83 to 110 (12%); cancer incidence was 15.5% with personalised thresholds. The higher incidence rates in that study compared to the findings in the present paper result from the secondary care setting, higher prior odds of cancer upon admission to hospital, and higher average age of tested patients (mean age 74.2 ± 14.3 years).

## 6. Implications for Practice

This study demonstrates that age- and sex-specific thresholds for platelet count support a nuanced approach to early diagnosis, targeting more at-risk groups. It also demonstrates the importance of studying local health datasets to ensure that the application of this approach is appropriate for the target population. Selecting a risk threshold at which patients are offered further investigation should be tailored to local guidelines, needs, and priorities. It is also important to consider what the subsequent clinical or diagnostic evaluation should be for these patients, especially in cases where there is no other clinical feature to point to a likely site of malignancy [18]. If the overall clinical picture makes cancer appear unlikely, it may well be appropriate to repeat the platelet count after an interval of four to six weeks. In England, a “raised platelet pathway” is in development, which will guide general practitioners on how and when to investigate patients with raised platelets. This pathway could be adapted and applied in other countries; further research would be needed to assess the likely impact on the healthcare system. Consideration also needs to be given to the practical application of personalised thresholds and how to incorporate these into blood test results, providing clinicians with an early alert to possible cancer in the absence of high-risk symptoms. While this study has demonstrated the potential of personalised thresholds for platelet count, it did not include estimates of whether cancer could be diagnosed earlier, in terms of days sooner or in terms of earlier disease stage if personalised thresholds were used. Further research is underway in our respective groups with audit data to quantify the potential patient benefit from this approach and whether personalised thresholds could result in diagnosis at an earlier disease stage. Also under investigation is what added value could accrue from considering platelet count in combination with other components of the full blood count, such as haemoglobin and other related indices [10], and how changes in test results over time could be of clinical use.

A raised platelet count is a non-specific feature of cancer since it suggests there is a risk of cancer but does not specify a specific cancer site. This is similar to weight loss, fatigue, and, to an extent, abdominal pain or anaemia. Multi-cancer early detection tests may be appropriate for raised platelet counts in the absence of other cancer symptoms, depending on their diagnostic performance in the symptomatic population [19]. As colorectal and lung cancers predominate in patients with raised platelets, initial investigation should target these sites. Further investigation, depending on the patient’s symptoms, could be CT imaging, endoscopy, or CA125. There may also be a role for platelet count in addition to existing screening protocols for high-risk populations.

## 7. Conclusions

Age- and sex-specific upper thresholds for platelet count could support cancer diagnosis for patients having a blood test in primary care. Studying national health datasets can ensure that any recommended personalised thresholds are locally appropriate. Further research is needed to assess the likely impact of new recommendations on the healthcare system and on cancer outcomes and patient benefit.

## Figures and Tables

**Figure 1 cancers-16-03074-f001:**
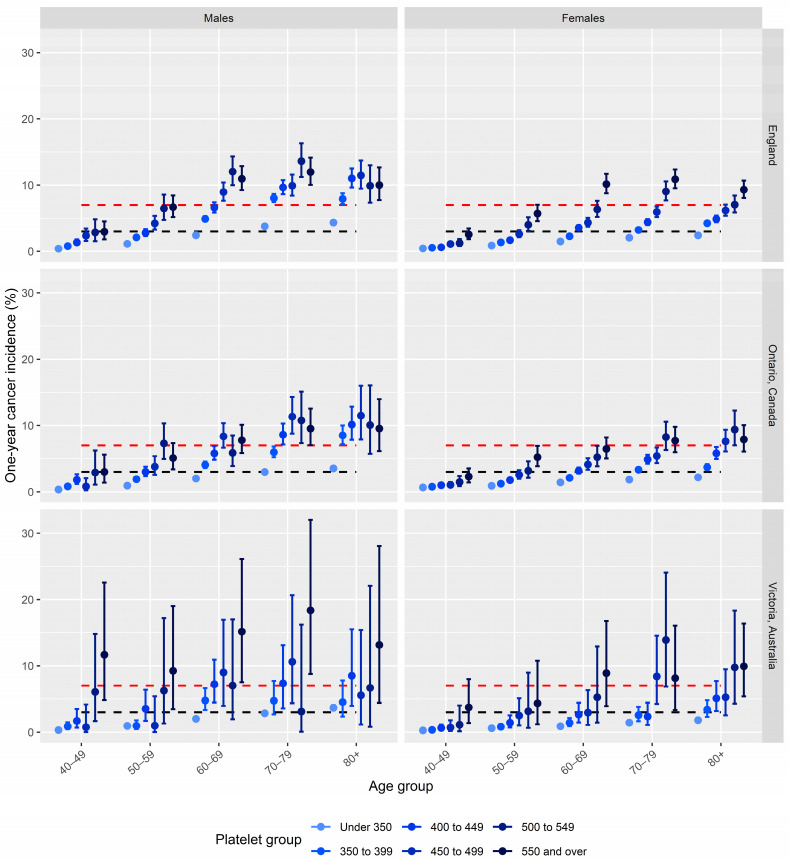
Cancer incidence following thrombocytosis In England, Canada, and Australia for female and male patients, stratified by age group and platelet count group. Black dashed link: 3% risk threshold. Red dashed line: 7% risk threshold.

**Table 1 cancers-16-03074-t001:** Patient demographics in the three cohorts.

	English Cohort N = 3,087,436	Canadian Cohort N = 3,140,451	Australian Cohort N = 266,900
Male sex, *n* (%)			
	1,366,827 (44.27)	1,392,750 (44.34)	117,690 (44.10)
Age, mean (SD)			
Men	62.6 (13.0)	60.8 (12.0)	57.3 (13.2)
Women	62.6 (14.3)	60.8 (13.0)	57.5 (14.4)
Age group, *n* (%)			
40 to 49	687,116 (22.26)	705,763 (22.47%)	96,502 (36.16%)
50 to 59	750,823 (24.32)	905,780 (28.84%)	65,865 (24.68%)
60 to 69	690,372 (22.36)	787,994 (25.09%)	51,560 (19.32%)
70 to 79	566,723 (18.36)	477,173 (15.19%)	29,492 (11.05%)
80 and over	392,402 (12.71)	263,741 (8.40%)	23,481 (8.80%)

## Data Availability

Data cannot be shared under the terms of the suppliers.

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
