# Peer review of "Improving the Clinical Utility of Platelet Count for Cancer Detection in Primary Care: A Cohort Study in England, Canada, and Australia"

_cancers, 2024, doi:10.3390/cancers16173074_

Round 1

Reviewer 1 Report

Comments and Suggestions for Authors

The review article by Mounce L. T. et al. have demonstrated that age- and sex-specific upper thresholds for platelet count that could support cancer diagnosis for patients having a blood test in primary care.

The manuscript is well written with clear description.  However, some concerns should be addressed.

1.         The title is so big, please make this title short and more focus for the researchers.

2.         Some of the previous studies already talked about the utility of platelet count in primary care in cancer patients, so how this current study is more relevant? Need a justification here.

3.         Authors need to show the flow chart of the population selection, identification, and analysis.

4.         In “patient demographics in the three cohorts” only male patients’ information is mentioned, author needs to add the female patient information as well in the table 1.

5.         Authors need to write the discussion part in a descriptive way with explanation.

6.         The interval time of references in the introduction is too long and contains a maximum of older references, so it is suggested to quote the literatures in the last three to five years. 

7.         Authors need to cite more references where they needed.

8.         The language expression of the manuscript needs to be further simplified and polished.

Comments on the Quality of English Language

Minor editing of English language required.

Author Response

Point 1: thank you for this feedback. We have amended the title to something more focussed: 

Improving the clinical utility of platelet count for cancer detection in primary care: a cohort study in England, Canada, and Australia. 

2. This study adds value over and above previous studies, as it explores how the platelet could be better used in primary care, if the patient's age and sex were considered in interpreting the test result. All previous studies to date have used a standard reference range for all participants, irrespective of age/sex. Our study is also novel as it is conducted in three countries; previous work has been limited to one setting. 

3. Thank you for this suggestion. We have added extra details on the number of patients excluded from each cohort due to having previous cancer. 

4. The values for age in Figure 1 are for all patients, irrespective of sex. We have only reported the number and percentage of one sex as is standard practice in scientific journals. 

5. We have ensured that the discussion begins with a descriptive summary of the findings, and that this is followed with an explanation of what we have described. 

6. The 'historical' references we have included are key seminal papers in this area of study, and are necessary to adequately frame the background to this work, and demonstrate how it has developed over time. We have included more recent papers on the link between platelet count and cancer detection, relevant to this study. 

7. Thank you, we have ensured that the manuscript is appropriately referenced throughout. 

8. We have reviewed the manuscript for content and language and have made some improvements. 

Reviewer 2 Report

Comments and Suggestions for Authors

Nice job!

The manuscript provides a useful information on the current understanding of platelet count and cancer.  The manuscript is very well written, and the introduction provides a good, generalized background of the topic that quickly gives the reader an appreciation of the applications of the technique. Certainly, this study will contribute much to the literature.

Thank you!

All the best

Candan HIZEL PERRY, PhD, HDR, MBA-HC

Consultant-expert in Genomics for Health Biotechnology & Personalized Medicine, Canada

Assoc. Director, Precision Medicine & Omics Platforms, CRCHUM - Université de Montréal, Montreal, QC, Canada

Director, R&D Pharmacogenetics, OPTI-THREA, Inc, Montreal, QC, Canada

Author Response

Thank you for your kind comments!

Reviewer 3 Report

Comments and Suggestions for Authors

The manuscript you have sent me to review is basically similar to a paper recently published

I honestly think this manuscript should have been rejected before sending for external review. The previously published paper is even more complete, providing more data than this manuscript.

Author Response

Thank you for your review. We believe this work to be novel as it is the largest every study of platelet count and cancer, with over 6,300,000 patients included from three countries. It is also the first to report age- and sex-stratified cancer incidence rates for patients by platelet count, and presents unique discussion of the clinical relevance of this work to primary care. 

Reviewer 4 Report

Comments and Suggestions for Authors

The article needs more refining.

Although thrombocytosis is an indirect marker for cancer, it would have been useful if the authors could provide details or expand the discussion section regarding which malignancy was most commonly associated with thrombocytosis

It is worth noting that no specific exclusion criteria were mentioned in the material and methods. Furthermore, it was stated that all types of malignancies, including hematological, were taken into consideration. it should be discussed if these could have influenced the results

Author Response

  1. Thank you for the suggestion. We have added this information to the results, and highlighted the most common sites and appropriate follow-up tests in the discussion.

2. Thank you - we have added the exclusion criteria and added further discussion about the sites of malignancy.